# Synthesis of Vapochromic Dyes Having Sensing Properties for Vapor Phase of Organic Solvents Used in Semiconductor Manufacturing Processes and Their Application to Textile-Based Sensors

**DOI:** 10.3390/s22124487

**Published:** 2022-06-14

**Authors:** Junheon Lee, Duyoung Kim, Taekyeong Kim

**Affiliations:** Department of Textile System Engineering, College of Engineering, Kyungpook National University, Daegu 41566, Korea; leon1013@hanmail.net (J.L.); endud0609@naver.com (D.K.)

**Keywords:** semiconductor, textile sensor, solvatochromism, vapochromism, dye

## Abstract

Two vapochromic dyes (DMx and DM) were synthesized to be used for textile-based sensors detecting the vapor phase of organic solvents. They were designed to show sensitive color change properties at a low concentration of vapors at room temperature. They were applied to cotton fabrics as a substrate of the textile-based sensors to examine their sensing properties for nine organic solvents frequently used in semiconductor manufacturing processes, such as trichloroethylene, dimethylacetamide, iso-propanol, methanol, n-hexane, ethylacetate, benzene, acetone, and hexamethyldisilazane. The textile sensor exhibited strong sensing properties of polar solvents rather than non-polar solvents. In particular, the detection of dimethylacetamide was the best, showing a color difference of 15.9 for DMx and 26.2 for DM under 300 ppm exposure. Even at the low concentration of 10 ppm of dimethylacetamide, the color change values reached 7.7 and 13.6, respectively, in an hour. The maximum absorption wavelength of the textile sensor was shifted from 580 nm to 550 nm for DMx and 550 nm to 540 nm for DM, respectively, due to dimethylacetamide exposure. The sensing mechanism was considered to depend on solvatochromism, the aggregational properties of the dyes and the adsorption amounts of the solvent vapors on the textile substrates to which the dyes were applied. Finally, the reusability of the textile sensor was tested for 10 cycles.

## 1. Introduction

Huge amounts of organic and inorganic solvents are being used in almost all industries and research. Most of them are generally recognized as hazardous and toxic to the human body as well as to the environment. In particular, volatile organic compounds have been attracting significant attention recently because they are easily vaporized even at room temperature to a dangerous level of concentration. Therefore, it is necessary to strictly monitor and reduce the atmospheric concentration, especially indoors. According to recent reports, organic solvents, which were used for semiconductor manufacturing processes, caused serious symptoms among workers, such as headaches, dizziness, confusion, nausea, dyspnea, hepatotoxicity, and central nervous system depression, and this has become a hotly debated social issue [1,2,3]. Various organic solvents are being used in semiconductor manufacturing processes, mainly washing and cleaning [4]. Although they are very carefully controlled, sometimes the leakage of solvents occurs inadvertently in the liquid or vapor phase. The leakage of the vapor phase is more serious because it may not be recognized or noticed easily in advance. Therefore, it is very useful and important for workers to perceive the leakage of the vapor phase of organic solvents with textile-based sensors without any electronic devices before the problem becomes serious.

The first role of dyes was the coloration of materials such as fibers, leathers, papers, plastics, and all kinds of substrates. In addition to this, the application of dyes to organic sensors increased recently [5,6,7,8,9,10,11,12,13,14]. They are generally known as chemosensors, which means that the dyes can detect target chemicals and physical stimuli. In order for dyes to be used as organic sensors, they must basically exhibit a color change after exposure to the chemicals or stimuli. This is called chromism and many instances are already well-known such as thermochromism, photochromism, solvatochromism, vapochromism, halochromism, and electrochromism [15,16,17,18,19,20]. As mentioned above, the detection of the vapor phase is more effective than that of liquid or solution phases. Unfortunately, the chromic properties of many sensor dyes are likely to be diminished or disappear inside textile substrates. This is because the stimulants may not penetrate into the substrates and, in some cases, the entrapment of sensing dyes inside the compact molecular structure of substrates may cause the restriction of structural deformation in the dyes, which is induced by the stimulants.

Many studies have reported that chromic dyes exhibit a color change in the powder state or the solution state, or otherwise, an extremely high concentration of vapor is necessary to induce a color change. In some cases, the color change was too weak to be recognized in daylight; therefore, experimental data were obtained under a UV light. 

In this study, two vapochromic dyes were synthesized to be used for textile-based sensors detecting the vapor phase of organic solvents used in the semiconductor manufacturing process (Figure 1). In order to exhibit effective vapochromism, the dyes need to be solvatochromic first, and then highly aggregative in nature. This is because aggregation of dyes can affect to the optical properties such as shift of absorption spectra [21]. After the exploration and investigation of many commercial direct dyes, we designed and synthesized two dyes showing strong color change upon exposure to the vapor phase of organic solvents even inside cellulose fibers at a low concentration at room temperature. They were applied to cotton fibers to fabricate textile-based vapor sensors and their sensing properties for nine organic solvents frequently used in semiconductor manufacturing processes were examined. Sensing mechanisms were investigated in terms of solvatochromism, the aggregative properties of the dyes, and the adsorption amounts of the solvent vapors in cotton fabrics. Considering practical application, reusability was also tested.

## 2. Materials and Methods

### 2.1. Materials

The following were used to synthesize the two novel dyes: 2-Amino-3,5-dimethylbenzenesulfonic acid, 2,5-dimethoxyaniline, 2,5-dimethylaniline, 7-anilino-4-hydroxy-2-naphthalenesulfonic acid, sodium hydroxide, sodium nitrite, acetic acid, and aqueous of hydrochloric acid. Trichloroethylene (TCE), N,N-dimethylacetamide (DMAc), iso-propanol (IPA), and methanol (MeOH), which are mainly used as cleaning solutions in the etching process, benzene, n-hexane, and ethylacetate as a photoresist, acetone as a photographic developer, and hexamethyldisilazane (HMDS) as a surface adhesive were used (S. Park et al., 2011). In addition, water, ethylene glycol, acetonitrile, dimethyl sufoxide (DMSO), pyridine, tetrahydrofuran (THF), and N,N-dimethylformamide (DMF) were used for other experiments. All reagents and solvents are commercially available and were used without further purification. A standard cotton fabric (ISO 105-F02, warp, 35 threads/cm; weft, 31 threads/cm; weight, 115 ± 5 g/m^2^) was used as a cellulosic substrate. Sodium sulfate was added in the dyeing process, and a practical polycationic fixing agent was used for the aftertreatment to improve the fastness of the dyed fabrics.

### 2.2. Synthesis of Dyes

The synthetic procedure of DMx (**5a**) and DM (**5b**) is shown in Figure 2. Both dyes were synthesized in two diazotization steps. In the first diazotization step, to substitute a sulfonic acid group to sodium sulfonate in the 2-amino-3,5-dimethylbenzenesulfonic acid (**1**), the 2-amino-3,5-dimethylbenzenesulfonic acid (**1**, 0.01 mol) was dissolved in an aqueous solution of sodium hydroxide (0.01 mol, 30 mL) and stirred at room temperature for 30 min and then sodium nitrite (0.01 mol) aqueous solution (5 mL) was added. The mixture was placed in an ice-water bath and the temperature was lowered to 0~5 °C. A hydrochloric acid (0.03 mol) aqueous solution was slowly added into the solution while maintaining the temperature at 0~5 °C and stirred for 1 h. As the diazonium reaction proceeded, the mixture solution turned into a yellowish opaque solution. After the diazonium reaction, the reactant was slowly added to 2,5-dimethoxyaniline (**2a**, 0.01 mol) as the first coupler dissolved in acetic acid (3 mL) while maintaining 0~5 °C. As the reactants underwent an azo coupling reaction, the color of the solution turned red. After stirring for an hour, the product was filtered and washed with deionized water several times. The product was dissolved in methanol (500 mL) and impurities were filtered out that were not dissolved in methanol. Then, the methanol within the product was partially evaporated to obtain a saturated solution, and an excess of water (500 mL) was poured to precipitate a monoazo compound (**3a**, yield: 15.3%) and dried under vacuum.

In the next diazotization step, a sulfonic acid group of the monoazo compound (**3a**, 0.001 mol) was substituted with sodium sulfonate in a sodium hydroxide (0.001 mol) aqueous solution (30 mL), and an aqueous solution of sodium nitrite (0.001 mol, 5 mL) was mixed, then an aqueous solution of hydrochloric acid (0.003 mol) was dropwise added while maintaining the temperature at 0~5 °C. The mixture solution was slowly added to the 7-anilino-4-hydroxy-2-naphthalenesulfonic acid (**4**, 0.001 mol) solution (10 mL), in which sulfonic acid was substituted with sodium sulfonate in an aqueous sodium hydroxide as performed previously, while maintaining 0~5 °C and pH 9~10 conditions within the sodium hydroxide (0.004 mol) aqueous solution (2 mL). The reaction was continued for an hour at that temperature, and sodium chloride (20 g) was added to help the precipitation of the synthesized dye (**5a**, DMx). After filtering, an excess of methanol (500 mL) was added to dissolve the dye and unreacted compounds except for sodium chloride, and filtering was performed again. To obtain a saturated solution, the methanol was evaporated, and then 1 mL of aqueous hydrochloric acid was added to substitute sodium sulfonate with sulfonic acid to diminish the ionization of the dye. An excess of water (500 mL) was added to help precipitate the bluish purple disazo compound, DMx (**5a**, yield, 18.6%), and dried under vacuum.

The synthesis procedure of **5b** (DM), a reddish violet compound, was the same as compound **5a** (DMx). Furthermore, 22.4% and 25.2% monoazo and disazo compounds were obtained, respectively.

### 2.3. Analysis of Dyes

In order to monitor the purity and the synthesis procedure, a high-performance liquid chromatography (HPLC) (Waters 510, column, C18, 5 μm, 4.6 mm × 150 mm; flow rate, 1.0 mL/min; ultraviolet detector spectrum, 254 nm, eluent for DMx (**5a**), 20 mM ammonium phosphate dibasic in 55% methanol and 45% water; and for DM (**5b**), 10 mM ammonium phosphate dibasic in 55% methanol and 45% water) was used. To verify the structure of the compounds, a liquid chromatography mass spectrometer (LC/MS) (XEVO-TQSmicro, Waters, Negative mode, ESI, capillary voltage, 3 kV; cone voltage, 40 V; flow rate, 0.2 mL/min; eluent, 50% methanol; and 50% water) and a proton nuclear magnetic resonance (NMR) spectrometer (Avance III 500, Bruker) were utilized. The NMR was analyzed in DMSO-d_6_ containing 0.03% TMS as an internal standard. To investigate the optical properties, an ultraviolet-visible spectrophotometer (Optizen 2010UV) was used. X-ray diffraction analysis was performed through Malvern Panalytical EMPYREAN operating in the reflection mode with Cu-K_α_ radiation (λ, 1.540598 nm, 2° ≤ 2θ ≤ 100°).

### 2.4. Solvatochromism

Solvatochromism was observed using nine solvents with different dielectric constants, an indicator of solvent polarity, such as water (78.3553), DMSO (46.8260), ethylene glycol (40.2450), DMAc (37.7810), acetonitrile (35.6880), MeOH (32.6130), acetone (20.4930), pyridine (12.9780), and THF (7.4257). The two dyes were dissolved in each solvent, and absorption spectra were measured.

For a deeper insight into the solvatochromic properties of the dyes, density function theory (DFT) calculation was performed. Geometry optimization was executed at the B3LYP/6-31+G(d,p) level. Electron transitions were also calculated using the time-dependent DFT method at the B3LYP/6-31+G(d,p) level and using the solvation model; the integral equation formalism variant polarizable continuum model (IEFPCM) for the nine solvents. From the calculation, the HOMO and LUMO energy values were obtained and compared with the experimental values of the maximum absorption wavelength of the dyes in each solution.

### 2.5. Application of the Synthesized Dyes to Cellulosic Fabrics

To optimize the dye concentration to exhibit the largest color change when exposed to the solvents, both dyes were applied at concentrations of 0.5~30% owf. A certain concentration of dye and 2.5 g of sodium sulfate were dissolved in 50 mL distilled water and stirred for 30 min at room temperature. Cotton fabrics (1.0 g) were dyed in the dyeing solutions at 100 °C for 1 h. After dyeing, the dyed samples were washed with water at room temperature five times. It is well-known that the mechanical and chemical properties of fiber do not change when dyed with a direct dye using the general dyeing process. The type of dyes used in this study were direct dyes and we used cotton fabric as the substrate for the sensor. Therefore, its physical properties are almost the same as the pristine cotton fabric.

### 2.6. Measurement of Color Strength and Change in Applied Dyes to the Cellulosic Fabrics

The color strengths and spectra of the textile-based sensors were measured using a color measurement instrument (*Spectrophotometer CM-3600d, Konica Minolta*) and expressed by *K*/*S* values obtained every 10 nm in the range of 360–740 nm using Equation (1) based on the reflectance (*R*) of a single wavelength. The measurement was performed using a 10º standard observer, and the illuminant was the standard light D_65_ [22,23].
(1)$$\frac{K}{S}=\frac{{\left(1-R\right)}^{2}}{2R}$$
where *K* is the absorption coefficient; *S* is the scattering coefficient; and *R* is the reflectance (0 ≤ *R* ≤ 1).

To measure a color change upon exposure to the vapor phase of the solvents, the concept of color difference (Δ*E*) was employed. There are some color formulas to estimate color differences. In this study, the CIELAB color space was utilized, which comprises *L**, *a**, and *b** factors. These three factors are plotted at three dimensions corresponding to the lightness (*L**), red to green (*a**), and yellow to blue (*b**) of the color vision. The color difference (Δ*E*) was calculated as follows (Equation (2)), where Δ indicates the difference before and after exposure to the vapor state of the solvents [24,25].
(2)$$\Delta E=\sqrt{{\left(\Delta L^{*}\right)}^{2}+{\left(\Delta a^{*}\right)}^{2}+{\left(\Delta b^{*}\right)}^{2}}\;\;\;\;$$

To examine the sensing properties of the cotton fabrics dyed with the two synthesized dyes, 2 mL of the nine solvents used in semiconductor processing were added to 20 mL vials, and the dyed fabrics (15 mm × 20 mm) were hanged into each vial under saturated vapor without any direct contact with the liquid phase of the solvents. The vials were sealed thoroughly with sealing film and left at room temperature for 24 h. The color difference of dyed samples before and after exposure was measured and compared. 

Three among the nine solvents, namely DMAc, MeOH, and acetone, were selected for quantitative experiments with respect to concentration (10 to 300 ppm) and time (10 to 200 min). The reason for selecting the three solvents is that they are included in the top four showing the strongest color change. In addition, even though the color change in the IPA is a little higher than acetone, considering the IPA is the same alcoholic solvent as MeOH it was not selected for these quantitative experiments. The internal volume of the vials was accurately measured, and then the three solvents of the corresponding concentration (10 to 300 ppm) were precisely measured and injected with a micro-syringe into the vials. In order to completely vaporize the solvents in the vials, the quantitative experiment was tested at 30 °C for acetone and MeOH, and 70 °C for DMAc. The maximum concentration, 300 ppm, of the VOCs was completely vaporized at those temperatures during experiments. 

### 2.7. Adsorption of Solvents on Cellulose Fabrics

The aim of this study was to detect solvents through a color change in the textile vapor sensors via the adsorption of solvent molecules. Therefore, the amount of solvents adsorbed by the substrate may have a significant impact on the sensing performance. The amounts of solvents adsorbed by the pristine cotton fabrics were analyzed using gas chromatography (GC-2030, Shimadzu, column; dimethylpolysiloxane, 30 m, 0.25 mm I.D., 0.25 μm, PerkinElmer, detector; FID). To obtain a calibration curve, various concentrations of the nine solvents were prepared, and the area of the gas chromatography was measured. The relationship between the adsorption amounts of the solvents on the fabrics and color differences was examined. An amount of 0.05 g of the pristine cotton fabrics was exposed to 300 ppm of vapor phase of the solvents at 30~70 °C for 200 min without direct contact, as mentioned above. After exposure, the fabrics were immersed in DMF (extraction solvent) to desorb the adsorbed solvents out of the fabrics. The adsorbed amounts of the solvents can be obtained accurately using the area of the solvents extracted from the fabrics via gas chromatography.

### 2.8. Reusability and Repeatability of Textile Sensors

The repeatability of the textile vapor sensors was examined. One sensor was exposed to 300 ppm of DMAc at 70 °C for 200 min, and the color difference was measured before and after exposure, and then it was kept inside a vacuum chamber for 24 h to desorb the adsorbed DMAc molecules completely out of the sensor fabric. After that, the process of exposure to DMAc, and the measurement of color change and desorption were repeated for 10 cycles.

### 2.9. Aftertreatment

The cotton fabrics dyed with the sensor dyes were treated with a conventional polycationic fixing agent to improve fastness. An amount of 5% of the polycationic fixing agent was used at 40 °C for 20 min (liquor ratio: 1:50). To confirm the color change property of the treated fabrics, the color difference was measured after exposure to 300 ppm of DMAc at 70 °C for 200 min and compared with the untreated fabrics. The durability against washing, rubbing, and light fastness were examined through the procedure of textile standard test methods, ISO 105-C06 A1S, ISO 105-X12, and ISO 105-B02, respectively.

## 3. Results and Discussion

### 3.1. Structural and Optical Analysis of the Dyes

As shown in Figure 2, there were two diazotization steps to synthesize DMx and DM. In DMx, the purity of monoazo (**3a**, first diazotization) and disazo products (**5a**, second diazotization) was verified using HPLC to be 90.7% and 94.2%, respectively. In DM, it was confirmed to be 92.7% and 96.4%. The structures of the dyes were analyzed using LC/MS and a proton NMR spectrometer. From the mass analysis of DMx, *m*/*z* for C_16_H_18_N_3_O_5_S^-^ (M-H)^−^ (monoazo compound) was expected to be 364.1 and found to be 364.0; for C_32_H_27_N_5_NaO_9_S_2_^−^ (M-Na)^−^ (disazo compound), *m*/*z* was expected to be 712.1 and found to be 712.0; for C_32_H_28_N_5_O_9_S_2_^−^ (M-2Na+H)^−^, *m*/*z* was expected to be 690.1 and found to be 690.5. In DM, *m*/*z* for C_16_H_18_N_3_O_3_S^−^ (M-H)^−^ (monoazo compound) was expected to be 332.11 and found to be 332.0; for C_32_H_27_N_5_NaO_7_S_2_^−^ (M-Na)^−^ (disazo compound), *m*/*z* was expected to be 680.11 and found to be 680.0; for C_32_H_28_N_5_O_7_S_2_^−^ (M-2Na+H)^−^, *m*/*z* was expected to be 658.11 and found to be 658.0. There were eight kinds of protons in the DMx monoazo compound (**3a**) that were observed in ^1^H-NMR (500MHz, DMSO-d_6_): 2.27 (s, 3H, CH_3_), 2.31 (s, 3H, CH_3_), 3.90 (s, 3H, CH_3_), 4.08 (s, 3H, CH_3_), 6.49 (s, 1H, ArH), 7.11 (s, 1H, ArH), 7.23 (s, 1H, ArH), and 7.47 (s, 1H, ArH). In the DMx disazo compound (**5a**), there were 13 kinds of protons in ^1^H-NMR (500MHz, DMSO-d_6_): 2.16 (s, 3H, CH_3_), 2.32 (s, 3H, CH_3_), 3.85 (s, 3H, CH_3_), 4.02 (s, 3H, CH_3_), 6.65 (s, 1H, ArH), 7.05 (tt, 1H, ArH, J = 7.3 Hz), 7.12 (dd, 2H, ArH, J = 8.8 Hz), 7.29 (dd, 2H, ArH, J = 8.9 Hz), 7.37 (t, 2H, ArH, J = 9.1 Hz), 7.47 (s, 1H, ArH), 7.54 (s, 1H, ArH), 7.56 (s, 2H, ArH), and 7.94 (s, 1H, ArH). In the DM monoazo compound (**3b**), there were eight kinds of protons: 2.15 (s, 3H, CH_3_), 2.20 (s, 3H, CH_3_), 2.31 (s, 3H, CH_3_), 2.33 (s, 3H, CH_3_), 7.03 (s, 1H, ArH), 7.22 (s, 1H, ArH), 7.29 (s, 1H, ArH), 7.53 (sd, 1H, ArH, J = 1.0 Hz). The DM disazo compound (**5b**) had 10 kinds of protons; 2.37 (s, 6H, CH_3_), 2.64 (s, 6H, CH_3_), 6.66 (s, 1H, ArH), 7.10 (m, 3H, ArH), 7.24 (d, 2H, ArH, J = 8.9 Hz), 7.30 (d, 2H, ArH, J = 9.0 Hz), 7.38 (s, 1H, ArH), 7.57 (s, 1H, ArH), 7.78 (s, 1H, ArH), and 7.98 (s, 2H, ArH).

The maximum absorption wavelengths of DMx and DM in water were 567 nm and 536 nm, respectively. DMx and DM are compounds with dimethoxy and dimethyl substituents, respectively; as their amounts of polar substituents increase, the maximum absorption wavelengths appear at a longer wavelength. The molar absorption coefficients of DMx and DM were 32,662 and 30,059, respectively.

### 3.2. Application of the Dyes to Cellulosic Fabrics

To fabricate a textile-based sensor exhibiting color change upon exposure to the vapor phase of solvents, the synthesized dyes were applied to standard cotton fabrics and the sensing properties were measured. To optimize the concentration of the dyes showing the maximum color change upon exposure to the solvents, the cellulosic fabrics were applied with various dye concentrations (0.5~30% owf) and then the dyed fabrics were exposed to the saturated vapor of DMAc. The color strength and color difference values, depending on the dye concentrations, are shown in Figure 3. As the dye concentration increased, the color strength increased and reached equilibrium at approximately 10% owf. However, the color differences before and after exposure showed a maximum value at 2% owf for DMx and 3% owf for DM, and then decreased as the dye concentration continued to increase. It is considered that the color of the sample was too light at the lower concentration and too dark at the higher concentration. Therefore, the optimal concentration of the dyes was determined to be 2% owf for DMx and 3% owf for DM.

### 3.3. Sensing Performance of the Dyes in Cellulosic Fabrics

The cellulosic fabrics (15 mm × 20 mm) dyed with 2% owf of DMx and 3% owf of DM were placed in a sealed vials containing saturated vapor of each solvent without any direct contact and exposed at room temperature for 24 h to observe sensing properties (color change), and the results are presented in Figure 4.

The two textile vapor sensors showed almost the same detecting behavior for the nine solvents. The color change was stronger for relatively polar solvents than non-polar solvents. It is thought that the dyes are polar compounds that are soluble in polar solvents, and the substrate is also a polar material, so it has stronger interaction to polar solvents. 

Among the nine solvents, three exhibiting a strong color change, namely, acetone, DMAc, and MeOH, were selected and to conduct quantitative experiments according to solvent concentrations (10~300 ppm) and exposure time (10~200 min). In order to evaporate completely in the vials, each test vial was heated up to 30 °C for acetone and MeOH, and 70 °C for DMAc. Although the test temperature was lower than their boiling points, it was enough to vaporize the 300 ppm of solvents. In Figure 5a, the color change increased with the increase in the concentration of the solvents and exposure time, and then reached equilibrium. There are several indicators of toxicology of solvents. In this study, the sensitivity and practicality of the sensors were investigated using the threshold limit value-time weighted average (TLV-TWA), which represents the maximum allowable exposure concentration during normal work, i.e., 8 h a day, 5 days a week [26,27,28]. The TLV-TWA of DMAc and MeOH were 10 ppm and 200 ppm, respectively, and the color differences were 7.7 and 5.4 for DMx and 13.6 and 9.9 for DM at the allowable exposure concentration within 1 h. Additionally, the TLV-TWA of acetone was 500 ppm; however, even at 300 ppm, the color difference value reached 4.6 and 7.3 in 1 h. Generally speaking, a color difference value of 1.0 or more is recognizable to the naked eye. This means that the dyes show strong sensing performance even inside a cellulose substrate under a very low concentration of solvents. Since the fabricated sensors of this study exhibited very high sensitivity even at a lower concentration than TLV-TWA, showing a color difference of much higher than 1.0, trace levels of solvent leakage can be detected and responded to instantly without an electronic device.

Figure 5b,c display the K/S spectral shift and change in *L**, *a**, and *b** values when the textile-based sensors were exposed to DMAc, which caused the strongest color change. The K/S spectral shift of all nine solvents exhibited the same trend. Upon exposure to the solvents, a hypsochromic shift (DMx, from 580 nm to 550 nm; DM, from 550 nm to 540 nm) and a hyperchromic shift occurred. The *L**, *a**, and *b** values for dyed fabrics with DMx were 31.22, 5.16, and −15.02 before exposure and 29.30, 18.23, and −6.10 after exposure. For DM, the values were from 37.03, 28.17, and −25.29 to 39.30, 46.66, and −6.80. This means that the textile-based sensors changed from bluish purple to reddish violet when exposed to solvents in the vapor phase.

To visualize the performance of the textile-based vapor sensor, we dyed cotton filament yarn with synthesized vapochromic dye (DM) and then embroidered it as the lettering “CHEMICALs” on a sheet of polyester fabric dyed with a conventional non-vapochromic disperse dye of a similar color to the DM dye, as depicted in Figure 6. The textile-based sensor was placed into a closed acrylic box containing 200 ppm of DMAc vapor for 200 min, and the textile-based sensor displayed the lettering (CHEMICALs) by changing color.

### 3.4. Sensing Mechanism: Solvatochromism

The detecting mechanism can depend on three factors: solvatochromism, the aggregative properties of dyes, and the adsorption amount of the solvents on cellulosic fabrics. Solvatochromism was measured by dissolving the synthesized dyes in various solvents such as water, DMSO, ethylene glycol, DMAc, acetonitrile, MeOH, acetone, pyridine, and THF. In order to study solvatochromism, the dyes must be dissolved completely in the solvents, and this is why the solvents of this experiment are partially different from those in Figure 4 showing the solvents frequently used in semiconductor manufacturing processes. The shift of the absorption spectrum by the polarity of the solvents is displayed in Figure 7a. The polarity was expressed by the dielectric constant, which was one of the indicators of the polarity of a material. Figure 7b shows the relationship between the dielectric constant of the solvents and the maximum absorption wavelengths. Both of the dyes showed similar patterns to positive solvatochromic dyes, which exhibited a bathochromic shift with increasing solvent polarity. 

The sensing mechanism of the textile vapor sensors was attributable to the solvatochromism of the dyes inside the fiber. Cotton fabrics as a substrate are hydrophilic material usually containing 7~10% water under standard conditions. This implies that the dye molecules are placed in water-rich environments inside the cellulose structure and exhibit a similar absorption spectrum to that of a dissolved state in the liquid phase of water, which is the most polar solvent compared with any other organic solvents. Upon exposure to less polar solvents than the water, the internal polarity of cellulose may be lowered, which brings about a hypsochromic shift in the absorption spectrum of the dyes (Figure 5b). Solvatochromism comes into view when a dye molecule is surrounded by solvent molecules. This situation can be realized even in substrates in which a dye molecule is closely surrounded by adsorbed solvent molecules, as well as substrate polymers. Therefore, the color change in this study is thought to be due to the solvatochromism inside the substrate containing water.

To observe the solvatochromic properties of the dyes from the molecular orbital energy point of view, a density functional theory (DFT) calculation was performed. After geometry optimization, the solvation effect was measured for the nine solvents used in the solvatochromism study. From the calculation, the Highest Occupied Molecular Orbital (HOMO) and Lowest Unoccupied Molecular Orbital (LUMO) energy was calculated and compared with the experimental results of the maximum absorption wavelengths of the dyes in each solvent. In Figure 8, the energy gap between the HOMO and LUMO of DMx and DM dyes became narrower as the dielectric constants increased. This is in good agreement with the fact that the maximum absorption wavelength of the dyes dissolved as the polar solvents shifted to a longer wavelength, as shown in Figure 7.

### 3.5. Sensing Mechanism: Aggregative Characteristics of the Dyes

The dyes of this study have highly aggregative properties because they are direct dyes. When exposed to solvents, solvent molecules permeate into the dye molecule aggregates and disassemble dye crystals, causing color change. To verify the change in the distance between the dye molecular planes due to exposure to solvents, an XRD analysis was performed. Figure 9 depicts the XRD patterns of DMx and DM before and after exposure to DMAc, which caused the strongest color change.

The XRD peak patterns of both of the dyes before and after exposure were very similar because the chromophore was the same and only the substituents in the middle of the molecular structures were different for dimethoxy groups and dimethyl groups. The dyes showed diffraction peaks at 27.3°, 31.7°, 45.5°, 53.9°, 56.4°, 66.2°, 75.3°, and 83.9° before exposure, and the molecular interplanar distance can be calculated using the Bragg Equation as 3.26 Å, 2.82 Å, 1.99 Å, 1.70 Å, 1.63 Å, 1.41 Å, 1.26 Å, and 1.15 Å, respectively. After exposure to DMAc, the intensity of the peaks significantly decreased. This indicates that the DMAc molecules partially broke the dye crystals or aggregates.

Most organic dyes are dissolved as monomeric states in good solvents but aggregates in poor or non-solvents. The aggregates are theoretically divided into two forms: J- and H-aggregates. According to the exciton theory, the excitonic state of the dye aggregate splits into two levels through the interaction of their transition dipoles. The molecules may form a head-to-tail arrangement (end-to-end stacking) in J-aggregates. The excitonic state in the molecule arrangements can be simplified into two levels through the interaction of their transition dipoles, such as unstable aggregation, exhibiting a higher transition energy due to charge repulsion and stable aggregation with a lower transition energy due to charge attraction compared with the monomeric state. In the H-aggregate, the molecules may aggregate in a parallel way (side-by-side stacking) and it also can be simplified into two arrangements. However, the unstable aggregative form in J-aggregates and the stable aggregative form in H-aggregates are forbidden transitions according to the exciton theory. Therefore, J-aggregates show the red shifted absorption and H-aggregates show the blue shifted absorption based on the free molecule (monomeric state) absorbance spectrum [29,30,31]. When the dyed cellulose fabrics were exposed to DMAc, the DMAc molecules adsorbed on the dye molecules aggregated inside the substrate, and then the dye molecules transitioned to a more monomeric state. This caused a hypsochromic and hyperchromic shift of the textile-based vapor sensors under DMAc exposure. From this, both dyes formed J-aggregates in the cellulose fabrics, and this was disrupted by the penetration of the vapor of solvents; finally, they changed to monomeric states. Therefore, the sensing behavior from the dye point of view was attributed to not only the solvatochromism but also the deformation of the dye aggregates.

### 3.6. Sensing Mechanism: Adsorption Amounts of Solvents on Cellulosic Fabrics

In relation to dyes, the color change mechanisms due to solvent exposure were solvatochromism and changes in aggregative characteristic. In order for these phenomena to occur, it is essential for the textile substrates to adsorb the solvent molecules. Therefore, the adsorption amounts of solvents on cellulose substrates and sensing performance are closely related. The relationship between the amounts of solvents adsorbed on cotton fabrics and color difference values before and after exposure to solvents in the vapor state is shown in Figure 10. The adsorption amounts of solvents into the substrates were linearly proportional to the color difference. This means that the greater the amounts of solvents adsorbed on the substrates, the stronger the color change.

In Figure 11, the relationship between the rate of adsorption of solvents on cellulose fabrics and the color change under 300 ppm of DMAc, MeOH, and acetone is presented. The behavior of the solvents adsorption and color change is very similar. The adsorption amount for the solvents and the color difference of the textile vapor sensors were closely related. 

In summary, three factors affect color change, namely solvatochromism, the aggregative property of dyes, and the adsorbed amount of solvents on fabrics. Among them, solvatochromism is the first major factor in the color change in the sensor, and the adsorption amount of solvents is considered to be the second factor to strengthen the color change in a similar way to the aggregation of dyes.

Both of the dyes showed similar sensing behavior, except for sensitivity. DM showed a much better sensing ability than DMx. This was thought to be due to the different substituents in the dye structure, such as dimethoxy groups for DMx and dimethyl groups for DM. The more hydrophobic substituents were introduced, the higher the sensing performance obtained. This means that this kind of difference in the structure cannot significantly affect the pattern or trend of color change.

### 3.7. Reusability and Repeatability

The textile vapor sensors fabricated in this study can be reused continuously. The color change property of the sensor can be maintained after repeated exposure to the vapor phase of solvents and ventilation cycles. This indicates that the color changed upon exposure to the solvents and returned to the original colors through the desorption of the solvents. This was because the solvatochromism disappeared, and the aggregates of the dye molecules were recovered as the solvents were desorbed. During 10 cycles, as displayed in Figure 12, the sensitivity was maintained at almost the same level as the initial sensing performance.

### 3.8. Aftertreatment

Although direct dyes have many advantages such as varieties of colors, reasonable price, and the simplicity of the dyeing process, they have poor washing fastness. The textile vapor sensors were employed using aftertreatment with a commercial polycationic fixing agent to improve the fastness. The durability of the dyed samples against washing, rubbing, and light was evaluated using the procedure of textile standard testing methods and is summarized in Table 1. The fastness ratings were improved by c one level. The color difference values of the treated textile-based sensors dyed with DMx and DM when exposed to 300 ppm of DMAc for 200 min were 13.2 and 22.2, respectively. The sensing ability retained approximately 80% or more compared with the untreated sensors.

## 4. Conclusions

The two solvatochromic dyes, DMx and DM, with planar structures to optimize aggregative properties were synthesized and they were dyed on a cellulosic substrate to fabricate the textile-based vapor sensors which detect low concentrations of semiconductor processing solvents in the vapor phase. The maximum absorption wavelength of the dyed cellulosic fabrics with the two dyes was 580 nm for DMx (bluish purple) and 550 nm (reddish violet). Upon exposure to the polar solvents, this changed to 550 nm and 540 nm, that is, the color changed to violet and red. DMAc was most effectively detected, and when 300 ppm DMAc was exposed for 200 min, the color difference values were 15.9 for DMx and 26.2 for DM. The sensitivity of DM was better than DMx; however, the sensing behavior was similar. The sensing performance was maintained after 10 repeat cycles. Aftertreatment was performed using a polycationic fixing agent to improve fastness, and sensing ability was kept at 80% or more.

The dyes in this study cannot be considered to have high selectivity against various organic solvents. However, they have different and high sensitivity against them instead. Although the highest allowance limit, such as TLV-TWA, was stipulated strictly by law, this does not mean that the solvents below the limit are safe for the human body; this just means that the limit values must be accepted practically. In this context, it is encouraging that the dyes of this study exhibited very high sensitivity at a lower concentration than TLV-TWA, showing ΔE of much higher than 1.0, which was the level of recognition by the naked eye. In addition, the organic solvents used in the semiconductor process, as listed in this study, were not used in the same facility all together; they were separated by distance. Therefore, the specific solvents only need to be monitored around specific facilities. Of course, if a dye has the highest level of selectivity, it is the ideal case in this kind of research. This requires further study.

## Figures and Tables

**Figure 1 sensors-22-04487-f001:**
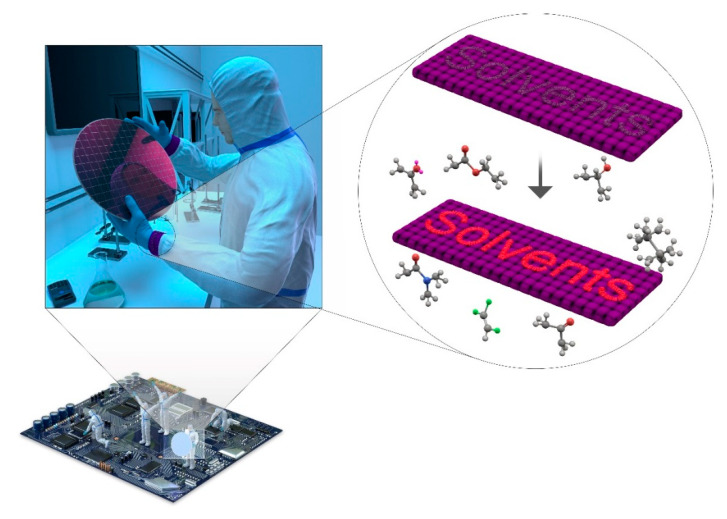
Schematic illustration of textile-based vapor sensor.

**Figure 2 sensors-22-04487-f002:**
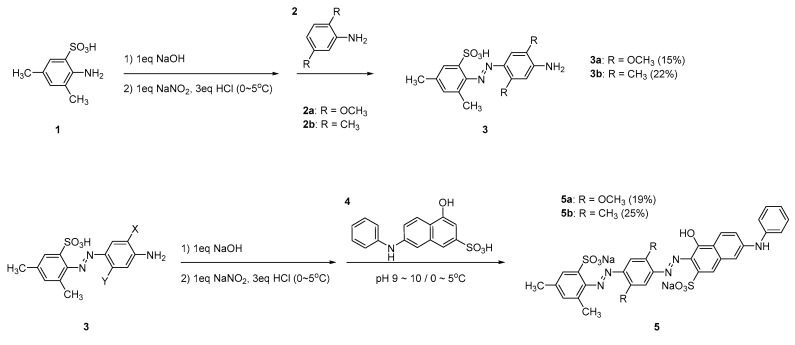
Synthesis procedure of two vapochromic dyes.

**Figure 3 sensors-22-04487-f003:**
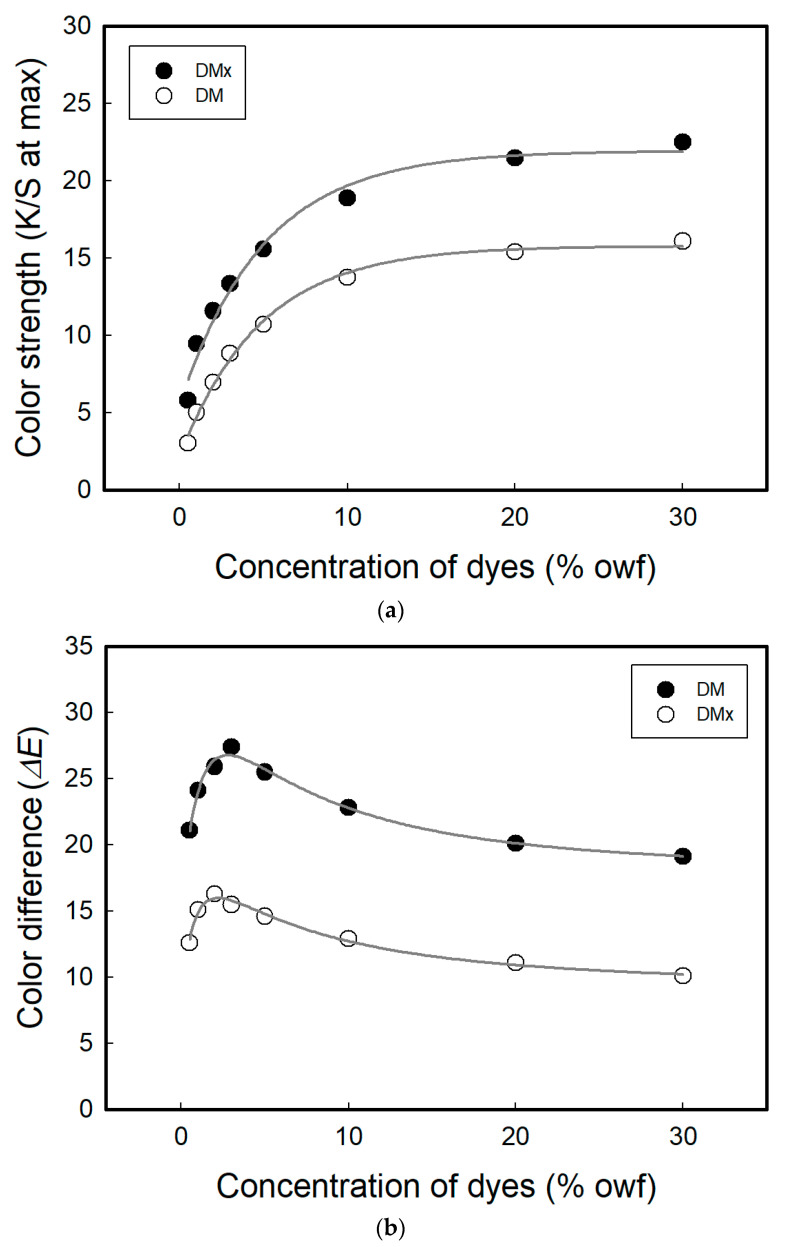
Color strengths at λ_max_ (**a**) and color differences (Δ*E*) in cellulosic fabrics dyed with DMx and DM before and after exposure to DMAc (**b**).

**Figure 4 sensors-22-04487-f004:**
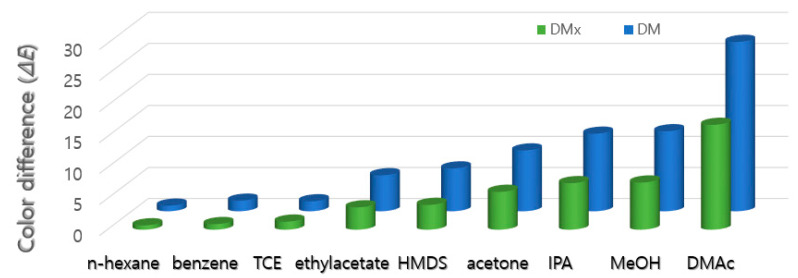
Color differences (Δ*E*) in cellulosic fabrics dyed with DMx and DM before and after exposure to semiconductor processing solvents.

**Figure 5 sensors-22-04487-f005:**
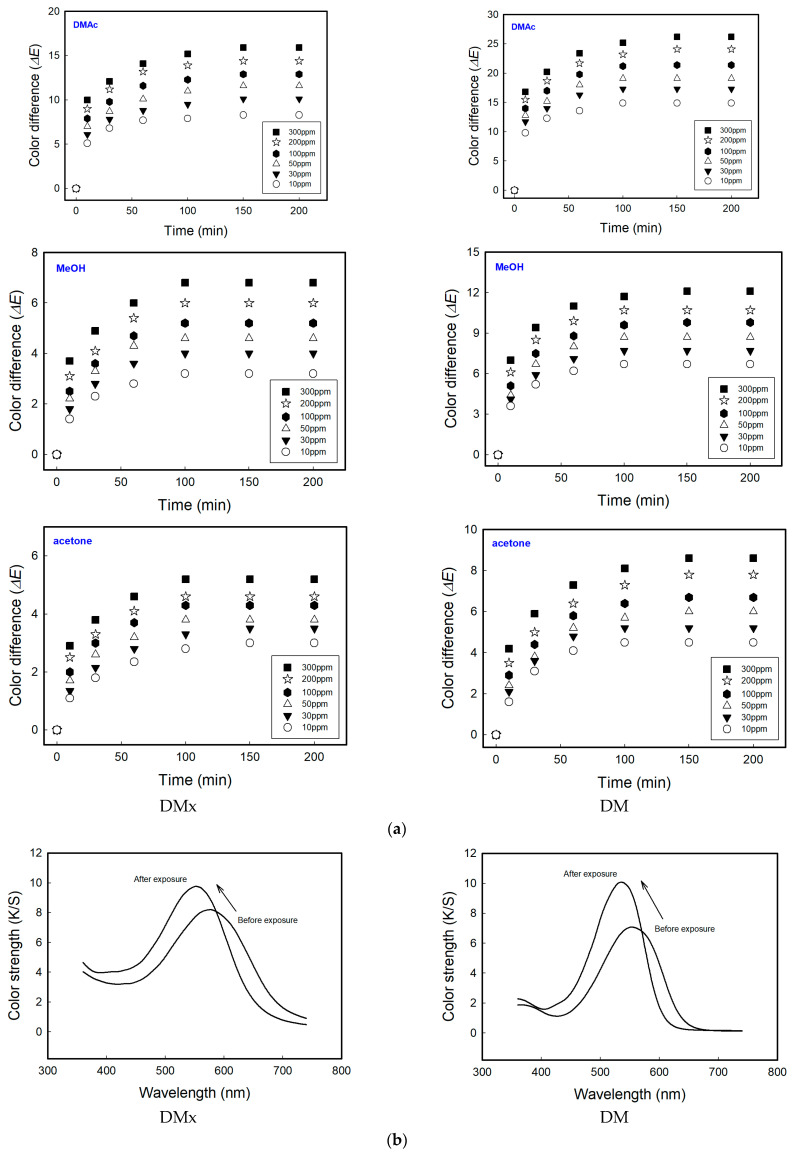
(**a**) Rate of color change in cellulosic fabrics dyed with DMx and DM upon exposure to DMAc, MeOH, and acetone. (**b**) Change in color spectra of dyed samples exposed to 300 ppm DMAc for 24 h. (**c**) Color coordination before and after exposure to 300 ppm DMAc.

**Figure 6 sensors-22-04487-f006:**
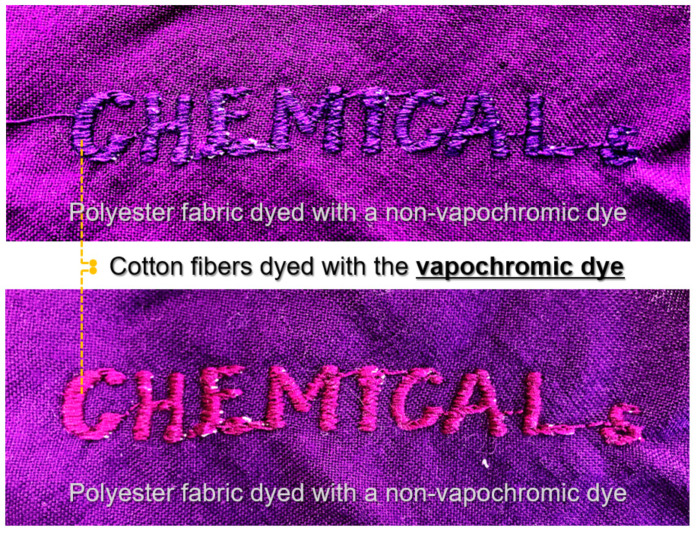
Visualization of the concept of textile-based vapor sensor.

**Figure 7 sensors-22-04487-f007:**
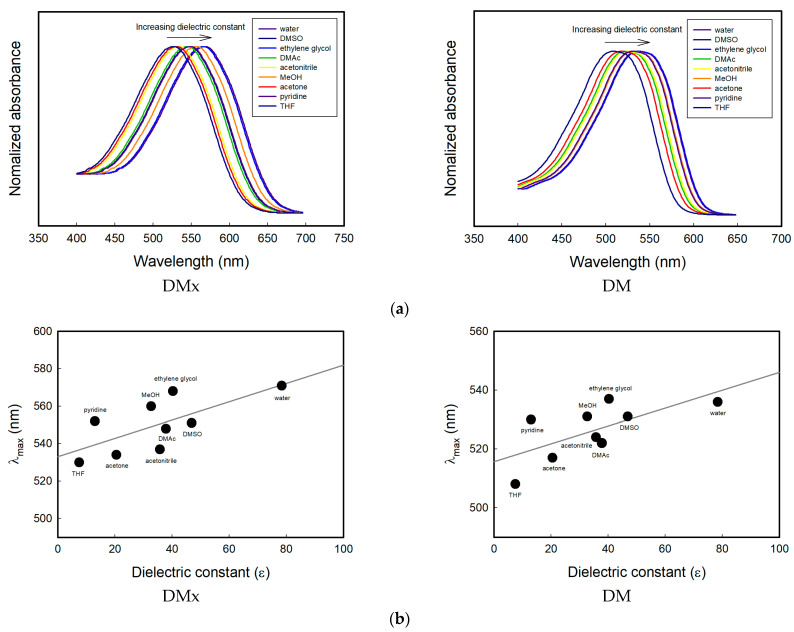
(**a**) Absorption spectra DMx and DM dissolved in various solvents with different dielectric constants, (**b**) relationship between the maximum absorption wavelength and solvent dielectric constant.

**Figure 8 sensors-22-04487-f008:**
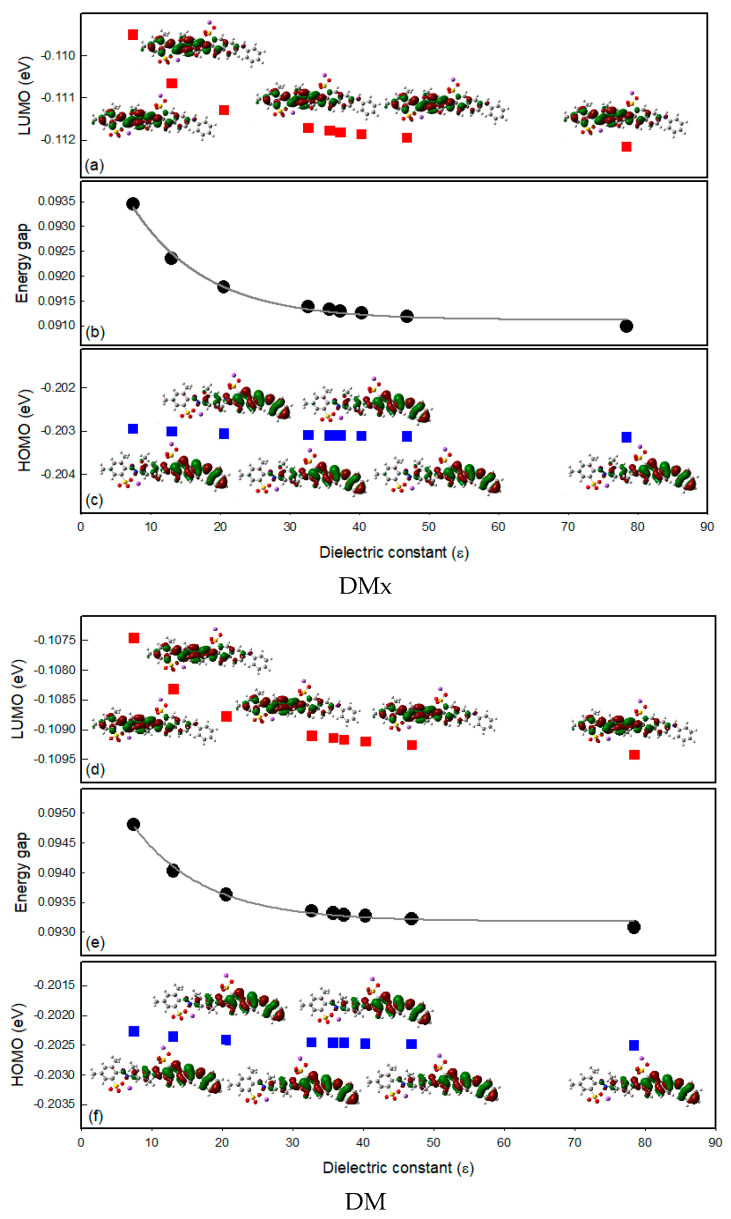
(**a**,**c**,**d**,**f**) Molecular orbital energy diagrams and isodensity surface plots of DMx and DM; (**b**,**e**) molecular orbital energy gap tendency for solvent dielectric constants.

**Figure 9 sensors-22-04487-f009:**
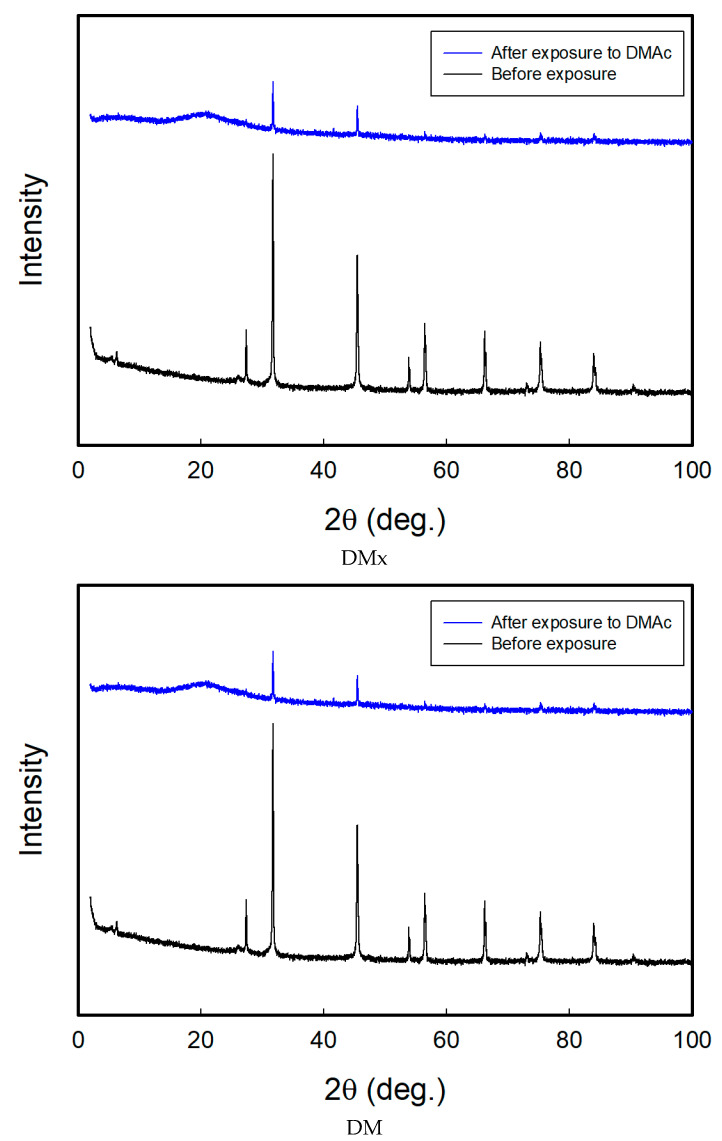
XRD patterns of DMx and DM before and after exposure to DMAc.

**Figure 10 sensors-22-04487-f010:**
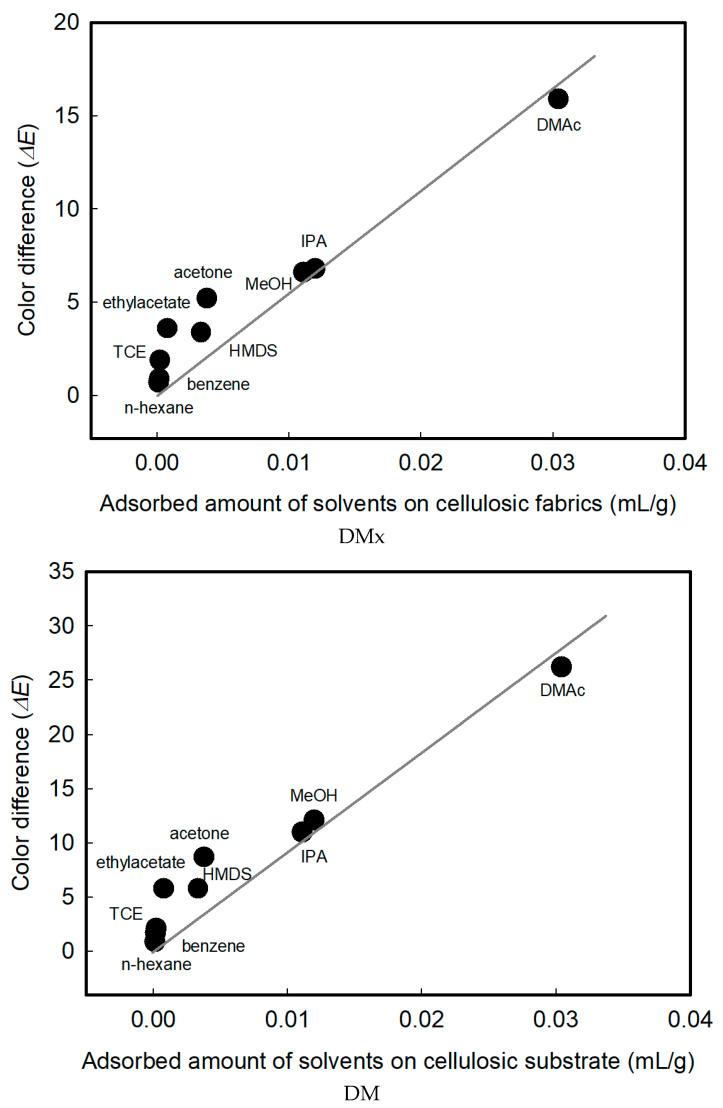
Relationship between the adsorption amounts of solvents on cellulosic fabrics and color difference before and after exposure to solvents.

**Figure 11 sensors-22-04487-f011:**
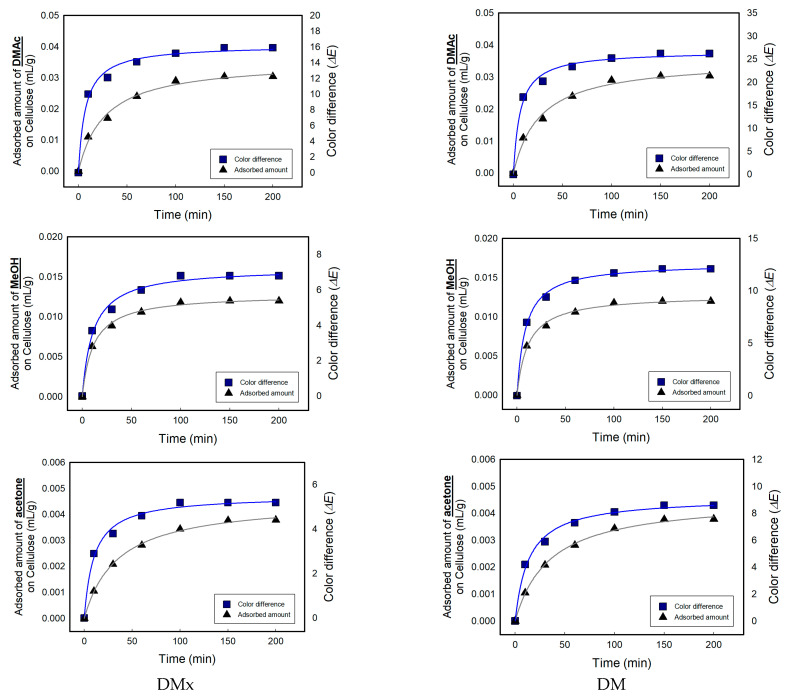
Relationship between the rate of adsorption of solvents on fabrics and color change.

**Figure 12 sensors-22-04487-f012:**
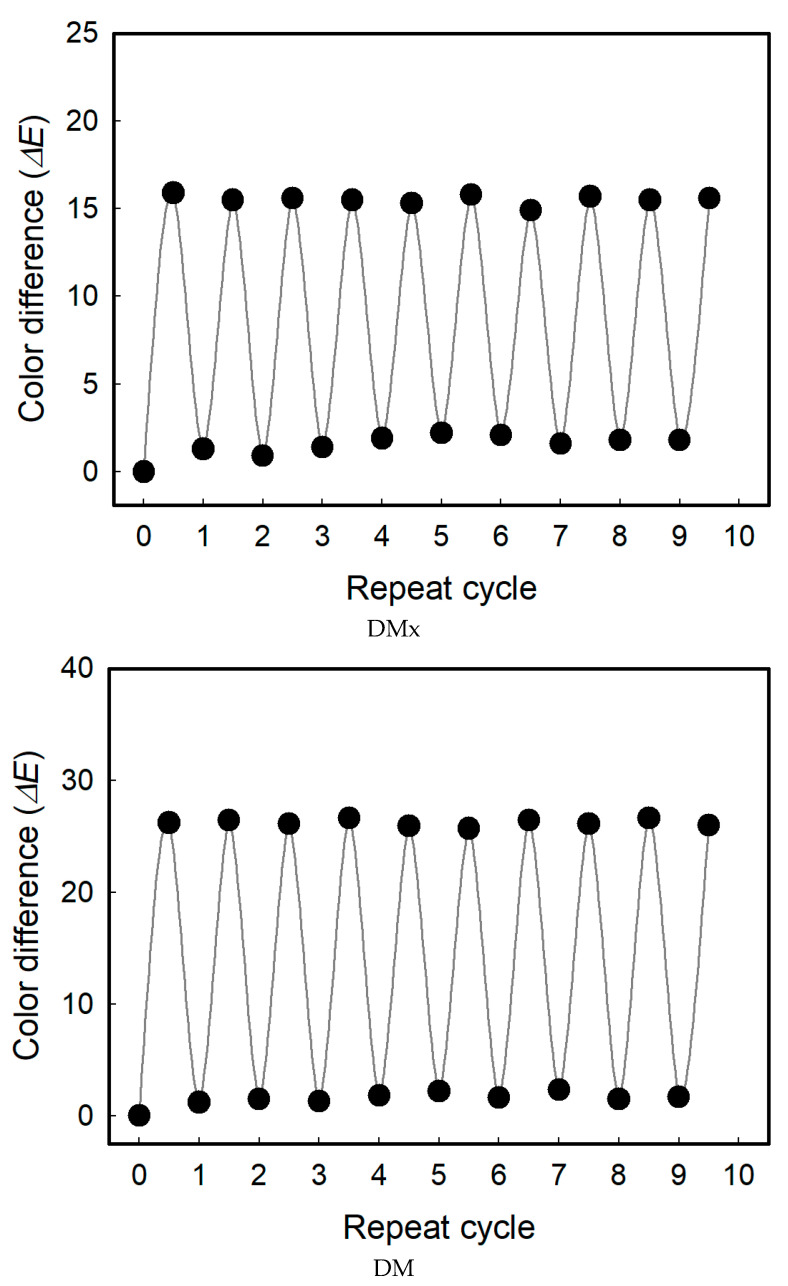
Reusability of the textile-based vapor sensors.

**Table 1 sensors-22-04487-t001:** Color fastness of untreated and treated textile-based sensors with a polycationic fixing agent.

Color Fastness			DMx		DM	
			Untreated	Treated	Untreated	Treated
Washing	Change in Color		3–4	4	3–4	4
	Staining	Acetate	3–4	4	3–4	4
		Cotton	1	2	1	2
		Nylon	3	4	3–4	4
		PET	4	4	4	4
		Acrylic	3–4	4	4	4
		Wool	3–4	4	3–4	4
Rubbing	Staining	Dry	4	4–5	4	4
		Wet	1–2	3	2–3	3–4
Light	Change in Color		3	4	3	4–5

## Data Availability

Not applicable.

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
