# Peer review of "Synthesis of Vapochromic Dyes Having Sensing Properties for Vapor Phase of Organic Solvents Used in Semiconductor Manufacturing Processes and Their Application to Textile-Based Sensors"

_sensors, 2022, doi:10.3390/s22124487_

Round 1

Reviewer 1 Report

In "Synthesis of Vapochromic Dyes Having Sensing Properties for Vapor Phase of Organic Solvents Used in Semiconductor Manufacturing Processes and Their Application to Textile-based Sensors", the authors explore the application of two dyes to be utilised as sensors to detect the chronical presence of solvent vapours in the working place. Two molecules were synthesised ( DM and DMx) and used as colourimetric indicators. Colour changes were estimated utilising differences in the Lab space. 

The manuscript's purpose is undoubted of great interest, and the materials are well characterised. However, in the reviewer's opinion, some points should be elucidated to fully understand the potentialities of materials based on the results obtained.

  • As the first concern, the response and sensitivity pattern of DM and DMx is strongly correlated to the amount of solvent absorbed in the textiles. This results in a high correlation between sensor patterns. The role of the indicator seems to be confined to signalling the presence of a solvent. Thus the solvatochromic effect and the characterisation in solution seem to be not correlated with the results obtained in the case of sensors. Furthermore, from the measurements reported in Figure 7, dyes seem to have very similar responses toward tested chemicals.
  • In this context, it would be interesting to see if changing the substrate for dyes is possible to change the response to the different chemicals.
  • In the actual condition, the sensors cannot differentiate the nature of the solvent. For example, let us suppose to have a ΔE of 10. It should trigger an alert in the case of acetone but not in the case of the other solvents (MeOH and DMAc). Usually, an "array" of sensors can overcome this problem if they have low correlated patterns of responses, and this is not the case. Thus the way to achieve the finalities of the manuscript should be carefully argumented. 
  • In the same context as the previous point, it is not clear how to estimate the threshold ΔE in a real context as suggested by the application mentioned by the authors. For example, while it is clear to the reviewer that small changes can be easily caught by sight, it is not clear how this change can be quantified accurately without a proper platform. In the reviewer's opinion, this point should be clarified as crucial for understanding how these sensors can be implemented or applied in real scenarios.
  • it is not clear to the reviewer what the dotted lines in Figure 5 stand for.
  • Please describe how the different vapour concentrations were provided.
  • to the reviewer, It is not clear the role of lightness in the ΔE formula. Lightness is usually related to illumination conditions and often is removed to reduce the noise since it is associated with external light conditions.

Author Response

As the first concern, the response and sensitivity pattern of DM and DMx is strongly correlated to the amount of solvent absorbed in the textiles. This results in a high correlation between sensor patterns. The role of the indicator seems to be confined to signalling the presence of a solvent. Thus the solvatochromic effect and the characterization in solution seem to be not correlated with the results obtained in the case of sensors. Furthermore, from the measurements reported in Figure 7, dyes seem to have very similar responses toward tested chemicals. In this context, it would be interesting to see if changing the substrate for dyes is possible to change the response to the different chemicals.

We really appreciate your deep and kind comments.

In our opinion, the sensing mechanism cannot be explained by just one factor. As already written in the manuscript, several factors affect color change such as solvatochromism, aggregative property of dyes, and adsorbed amounts of solvents on fabric. If only the adsorption amounts of solvents were the single major factor for color change of the sensor, the benzene or n-hexane, which were the solvents showing actually no color change in this study, would exhibit very strong effect on color change when applied in super-saturated environment like produced by boiling the solvents. In order to clarify this, we tested the situation after taking the reviewer’s comments. According to the results of that, the textile sensor actually did not show any color change even in super-saturated vapor condition produced by boiling benzene and n-hexane which makes the cotton fabric almost wet. It means that the major factor of color change of the sensor is thought to be solvatochromism first rather than adsorption amounts of solvents. The solvatochromism comes into view when a dye molecule is surrounded by solvent molecules. This situation can be realized even in substrates in which a dye molecule is closely surrounded by adsorbed solvent molecules as well as substrate polymers. Therefore, the effect of adsorption amount is considered to be the second effect to strengthen the color change as similar as aggregation of dyes (Inserted into “Sensing mechanism: Solvatochromism” and “Sensing mechanism: Adsorption amounts of solvents on cellulosic fabrics” in Result and discussion).

The two dyes, DM and DMx, have similar structure except for central part of the chromophore. One of our purposes of this study was to investigate the effect of chemical structure of dye molecules on vapochromism. As shown in many Figures, the absolute values of color change were different almost double between DM and DMx. However, the patterns or trend of color change against various solvents appeared almost same in the most experiments and this is in good agreement to energy calculation showing simulated solvatochromism. It means that this kind of difference in the structure cannot affect mainly the pattern or trend of color change against organic solvents (Inserted into Conclusion).

Since the dyes used in this study can be classified into direct dyes which have affinity toward polyamide (nylon) and wool as well as cellulose(cotton). As you kindly commented, we applied the dyes to nylon and wool to examine the sensing properties to organic solvents. However, the sensing property on nylon and wool was not strong compared to those on cotton fiber, showing very slight color change. In fact, the visual color of the sensor dyes on nylon and wool was relatively reddish rather than purple violet on cotton. These dyes exhibit sensing performance by shift of absorption spectrum to shorter wavelength (red) from longer wavelength (purple violet). On cotton fabric, the shift from purple violet to red after exposure to solvents brings about significant color change visually and instrumentally. However, on nylon and wool, the shift of absorption spectrum was not recognized because the color shade was the same red before and after exposure.

In the actual condition, the sensors cannot differentiate the nature of the solvent. For example, let us suppose to have a ΔE of 10. It should trigger an alert in the case of acetone but not in the case of the other solvents (MeOH and DMAc). Usually, an "array" of sensors can overcome this problem if they have low correlated patterns of responses, and this is not the case. Thus the way to achieve the finalities of the manuscript should be carefully argumented. 

In the same context as the previous point, it is not clear how to estimate the threshold ΔE in a real context as suggested by the application mentioned by the authors. For example, while it is clear to the reviewer that small changes can be easily caught by sight, it is not clear how this change can be quantified accurately without a proper platform. In the reviewer's opinion, this point should be clarified as crucial for understanding how these sensors can be implemented or applied in real scenarios.

As you correctly commented, the dyes of this study could not be considered to have high selectivity against various organic solvents. However, they have different and high sensitivity against them instead. As explained in the manuscript, the highest allowance limit such as TLV-TWA was stipulated strictly in law. This means that the limit values are forced to be accepted practically and does not mean that the solvents below the limit are safe to human body. Therefore, if sensor dyes could react even very low concentration, it would be more useful. In this context, the dyes of this study exhibited very high sensitivity at the lower concentration than TLV-TWA showing ΔE of much higher than 1.0 which was the level of recognition by naked eyes (Inserted into “Sensing performance of the dyes in cellulosic fabrics” in Result and discussion and Conclusion).

In addition, the organic solvents used in semiconductor process as listed in this study are not used in the same facility altogether. They are separated in space at a distance. Therefore, practically the specific solvents only need to be monitored around specific facilities. Of course, if a dye should have the highest level of selectivity, it will be the ideal case in this kind of research. However, it is hard for us to think that this is possible at this moment (Inserted into “Sensing performance of the dyes in cellulosic fabrics” in Result and discussion and Conclusion).

The idea of “array” is a good option to this kind of research, and we have been thinking about the application to our study. In order to realize the idea, various sensor dyes to have different sensitivity against solvents are necessary, and we keep trying to synthesize various sensor dyes. At this moment, several novel dyes besides ones of this study are being tested in our research group. If possible, the results of the experiments using the “array” sensors will be reported later.

The objective of this study is to fabricate the textile-based sensors that detect solvents without any electronic device. Since the color change of the sensors of this study could be easily recognized by naked eyes even at very low concentration of organic solvents below TLV-TWA, we thought that there is no need to estimate threshold ΔE by a separate device platform. If necessary, the estimation of threshold ΔE is possible by combination of the sensor dyes and practically available tiny electronic color sensors. Apart from this study, the hybrid sensor was investigated in our research group. If we have useful academic results, we will report later (Inserted into Introduction and “Sensing performance of the dyes in cellulosic fabrics” in Result and discussion).

It is not clear to the reviewer what the dotted lines in Figure 5 stand for.

I really appreciate your detailed comment. I inserted the dotted lines to emphasize TLV-TWA, but it might cause misunderstanding by insertion the dotted lines. So I deleted the lines in all figures (Figure 3 (a)) and it is explained in text in “Sensing performance of the dyes in cellulosic fabrics” part.

Please describe how the different vapour concentrations were provided.

The internal volume of the vial was accurately measured, and then the solvent of the corresponding concentration was precisely measured and injected with a microsyringe. The test temperature was slightly elevated (MeOH, acetone in 30°C and DMAc in 70°C) to vaporize of them completely (Supplement to “Measurement of color strength and change of applied dyes to the cellulosic fabrics” in Matrials and methods).

To the reviewer, It is not clear the role of lightness in the ΔE formula. Lightness is usually related to illumination conditions and often is removed to reduce the noise since it is associated with external light conditions.

As you know, the color difference equation has been scientifically established to calculate using all three elements of L*, a*, and b*, so it seems that the color difference cannot be calculated without L*. In addition, a standardized color measurement instrument was used so that it was not affected by the external illumination light.

Reviewer 2 Report

I think this is a quite nice paper for textile vapor sensors. Questions: 1. How about the stretchability of the sensor? 2. Will the dyes coating affect the breathability and softness of the origin textile? 3. Is there any benchmark to compare the design and the state of the art design?

The amended one can be considered for publication.

Author Response

We really appreciate your deep and kind comments.

How about the stretchability of the sensor?

It is well known that the mechanical and chemical properties of the fiber do not change when dyed with a direct dye using the general dyeing process. The type of dyes used in this study is direct dyes and we used cotton fabric as the substrate for the sensor. Therefore, its stretchability is almost the same as the pristine cotton fabric (Inserted into “Application of the synthesized dyes to cellulosic fabrics” in Materials and methods).

Will the dyes coating affect the breathability and softness of the origin textile?

As described above, the fabric is not modified at all by the process of this study, so the application of dyes does not affect the breathability and softness of the cotton fabric (Inserted into “Application of the synthesized dyes to cellulosic fabrics” in Materials and methods).

Is there any benchmark to compare the design and the state of the art design?

We searched for many previous studies in the beginning. Most of the research reported the color change properties of the dye themselves in the powder state. Even when applied to substrates, most of them presented the results after exposure to very high concentration of solvent vapor. In some cases, the color change was so weak to be recognized under the daylight, therefore experimental data was obtained under UV light. In contrast to the reported researches, our sensor dyes of this study show high sensitivity and visual recognition under the daylight at room temperature (Inserted in Introduction part ).

In order to strengthen the visual recognition, we dyed cotton filament yarn with the sensor dyes and then embroidered the filament to display CHEMICALs on polyester fabric dyed with non-vapochromic disperse dyes of the same color and depth, as depicted in Figure 6. This is thought to be one of unique points of this study.

Round 2

Reviewer 1 Report

The authors satisfactorily commented the reviewer's remarks and edited the manuscript that is now worth being published in its present form.

Author Response

Thank you very much for this comment as well as the previous comments. And I requested the professional native correction service by MDPI to edit the manuscript with correct grammars and terms.

I appreciate your reviews again and if you have any question or comments, please don't hesitate.

sincerely,

Junheon Lee, Duyoung Kim, and Taekyeong Kim
